# Incorporating the Results of Geological Disaster Ecological Risk Assessment into Spatial Policies for Ecological Functional Areas: Practice in the Qilian Mountains of China

**Xu Long, Qing Xiang, Rongguang Zhang * and Hong Huang**

Chengdu University of Technology, Chengdu 610059, China; longxu2020@126.com (X.L.);
xiangqing@stu.cdut.edu.cn (Q.X.); huanghong@stu.cdut.edu.cn (H.H.)
* Correspondence: zhangrg6880@163.com

**Abstract:** Geological hazards cause changes in the quality of the ecological environment, affect the function and stability of ecosystems, and negatively impact the maintenance and restoration of ecological functions in ecological functional areas (EFAs). This study integrates machine learning, geographic information technology, and multivariate statistical analysis modeling to develop a technical framework for quantitative analysis of ecological risk assessment (ERA) based on the causal logic between geological hazards and ecosystems. The results of the geological disaster ERA are mapped to EFAs, effectively identifying and quantifying the risk characteristics of different EFAs. The results show that: (1) The hazard–vulnerability–exposure ERA framework effectively identifies the distribution characteristics of high ecological risk around the Qilian Mountains, with high risk in the east and low risk in the west. (2) In high ecological risk areas, high hazard–high vulnerability–low exposure is the main combination pattern, accounting for 83.3%. (3) Overall, hazard and vulnerability have a greater impact on geological disaster ecological risk than exposure, with path coefficients of 0.802 (significant at $p = 0.01$ level) and 0.438 (significant at $p = 0.05$ level), respectively, in SEM. The random forest model ($R^2 = 0.748$) shows that social factors such as human density and road density contribute significantly more to extreme high risk than other factors, with a contribution rate of up to 44%. (4) Thirty-five ecological functional units were systematically grouped into four clusters and used to formulate a "layered" spatial policy for EFAs. The results of the research are expected to provide support for maximizing the policy impact of EFAs and formulating management decisions that serve ecological protection.

**Keywords:** geological disasters; ecological risk assessment (ERA); machine learning; ecological function areas (EFAs); spatial planning

## 1. Introduction

Ecological risk assessment (ERA) is a methodology for assessing and predicting the extent of ecosystem loss and degradation caused by human activities or natural hazards, in order to determine the state of ecosystem quality and the level of risk [1–3]. ERA provides an integrated assessment of human activities and environmental systems by identifying and linking the possible impacts of risks over long time periods and large regional scales in environmental management [4–6]. Ultimately, ecological environmental risk management and environmental monitoring provide decision support. ERA of geohazards is one of the hotspots of current research [7–9], focusing on the impacts and extent of damage to the ecological environment caused by geohazards [10–13] and the impacts and threats to ecosystem stability and function [14–16]. The main research contents of geohazard ERA include hazard assessment, exposure assessment, receptor analysis, and risk characterization [17–20]. ERA of geohazards is mostly performed by selecting multiple risk sources [21–23] and using ecological index and vulnerability index as assessment indicators [24–27], but studies on ERA of multiple risk sources at landscape scale are still relatively rare.

Ecological function areas (EFAs) is a type of geospatial zoning based on ecosystem characteristics, stress processes and effects, the importance of ecological service functions, and ecological sensitivity [28]. It is the basis and prerequisite for the implementation of regional ecological environment zoning management, which aims to clarify the important areas of regional ecological security and key areas for protection, analyze the existence of ecological problems and vulnerable areas, and provide a scientific basis for ecological protection and construction planning. It is the basis and prerequisite for the implementation of regional ecological environment zoning management [29]. There is a close relationship between ERA and EFAs, where the receptor of ERA is usually the ecosystem service function, and EFAs are areas whose main function is to provide ecological products [30,31]. We found that current research results on ERA focus more on the exposure of ecosystem services as a whole, and lack the differentiation of ecosystem services in terms of type and space, which makes it difficult to use the risk assessment results in the management and policy formulation of EFAs.

The Qilian Mountain region is important for China's ecological security and protection. However, there are prominent problems in the maintenance and restoration of the regional ecological function of the Qilian Mountains due to geological disasters and anthropogenic factors. Therefore, there is a current need to understand how to more effectively use the EFAs for policy impact and the development of services for ecological protection and management to protect the stability of the ecosystem services in the Qilian Mountain ecosystem.

Therefore, the objectives of this study are: (1) to develop a framework for geological disaster ERA; (2) to analyze the spatial distribution, patterns, and mechanisms of geological disaster ecological risk in the Qilian Mountains region of China; (3) to integrate the results of geological disaster ERA into policies for EFAs. The aim is to provide scientific references for the ecological protection and management of the Qilian Mountains region.

## 2. Materials and Methods

### 2.1. Study Area

In this study, Qilian Mountain was selected as the study area, the region is one of China's national nature reserves and is a world-class wetland ecological reserve [32] (Figure 1), which is an important place for plateau birds and fowl to live and reproduce, and at the same time, it is the only habitat for the endangered Przewalski's gazelle [33]. On the other hand, various geological disasters such as landslides and rockfalls frequently hit the area [34,35], posing a serious threat to the ecological security of the region. The Qilian Mountains serve as an important ecological barrier and water conservation area in the northern Qinghai–Tibet Plateau [36,37]. For the ecological protection and restoration of the ecological barrier area in northern Qinghai, it is urgent to propose spatial policies to address the ecological risks of geological hazards and curb the trend of ecological degradation [38,39].

### 2.2. Data Sources and Data Pre-Processing

The main data sources in the research area are (1) compilation maps of historical collapses and landslides, as well as relevant field survey data; (2) digital elevation model (DEM) obtained from the Geographic Spatial Data Cloud (http://www.gscloud.cn/ (accessed on 10 January 2022)), mainly used for acquiring terrain and landform basic environmental factors such as elevation, slope, and aspect; (3) rock type data obtained from the Chinese Academy of Sciences Resource and Environment Science Data Center (http://www.resdc.cn (accessed on 10 January 2022)); (4) Landsat-8 remote sensing imagery used to obtain land use data and the normalized difference vegetation index (NDVI); (5) data on rivers, roads, population density, and fault zones obtained from the National Geographic Information Resource Catalog Service System (https://www.webmap.cn (accessed on 10 January 2022)).

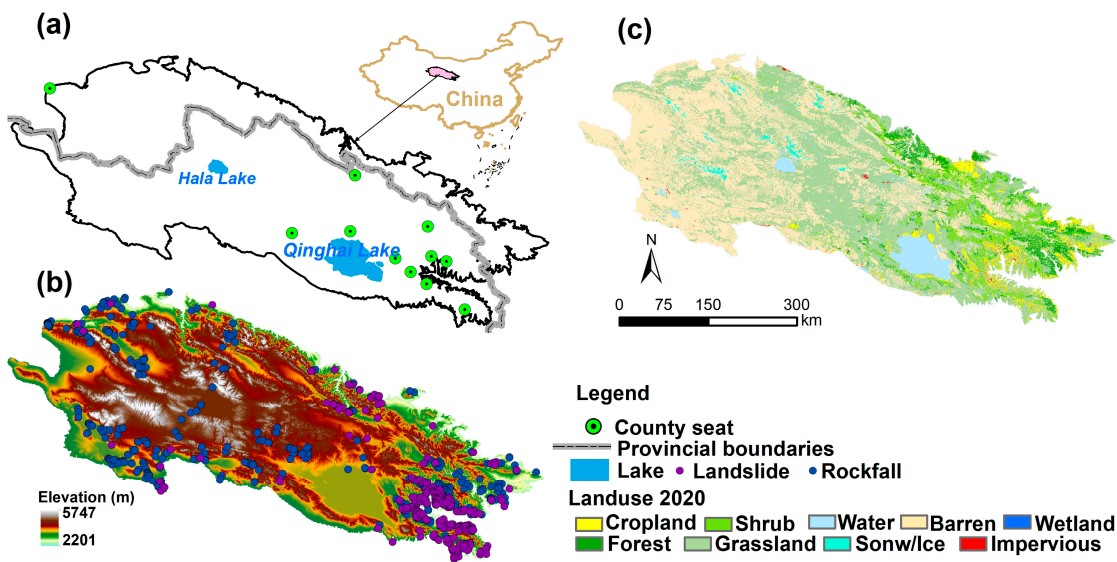

**Figure 1.** Overview of the study area: (**a**) location; (**b**) distribution of landslides and rockfall; (**c**) land use in 2020.

## 2.3. Methodology

Based on the causal logic between external disturbances and ecosystems, a framework for ERA was constructed (Figure 2). Hazard reflects the probability and severity of geological hazards, while vulnerability reflects the ability of landscape patterns to withstand geological hazards. Exposure reflects the sensitivity of ecosystem functions to geological hazards. Based on the evaluation of hazards, vulnerability, and ecosystem function values, the calculation formulas are as follows [40]:

$$ER = H \times V \times E$$

where *ER* is ecological risk, *H* is hazard, *V* is vulnerability, and *E* is exposure.

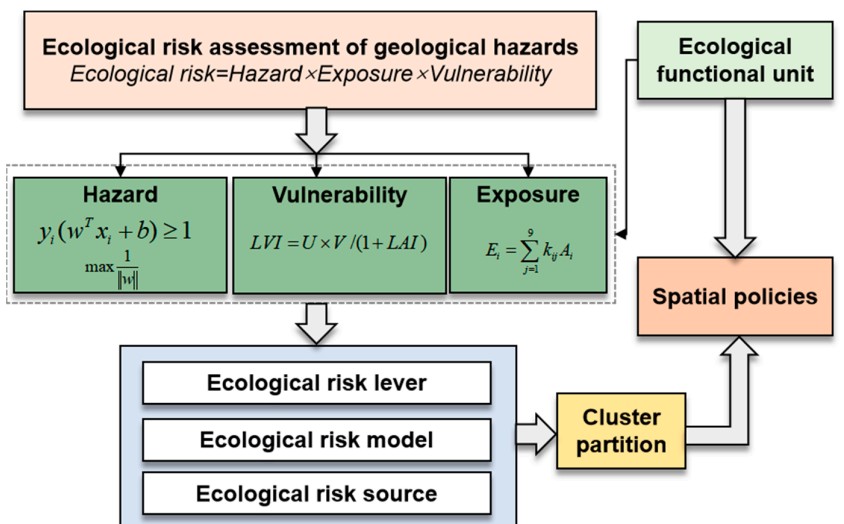

**Figure 2.** The technical process of incorporating the results of geological disaster ERA into policies for EFAs.

### 2.3.1. Hazard Calculation

Step 1: First, we selected the common driving factors for different subclasses of geological hazards, including elevation, slope, aspect, terrain wetness index, lithology,

soil type, fault, annual precipitation, temperature, NDVI, roads, population density, and 12 other factors (Figure 3). After resampling, the spatial resolution of all factor data was set to 1000 m. Different factor layers were stacked in a specific order to form multi-channel raster data.

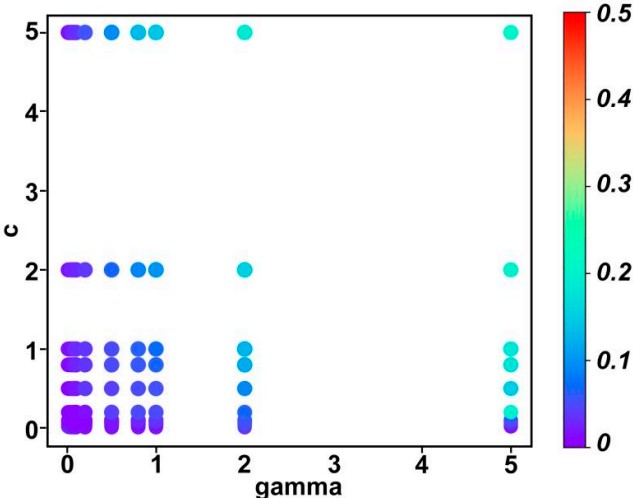

**Figure 3.** SVM parameter gamma and c selection.

Step 2: Random non-geological hazard points were generated outside the 5–10 km buffer distance from geological hazard points to construct data point samples. Geological hazard and non-geological hazard block datasets were then created based on the multi-channel raster data overlaid in Step 1. In addition, the training and test samples were split in a 7:3 ratio.

Step 3: The entire sample set was fed into the SVM model. The radial basis function (polynomial kernel) was used to evaluate the risk of geological hazards. The grid search algorithm was used to optimize the parameters and find the optimal gamma and penalty factor C [41]. The SVM formula is as follows:

$$y_i(w^T x_i + b) \geq 1$$

$$\max \frac{1}{\|w\|}$$

Step 4: The receiver operating characteristic (ROC) curve is a comprehensive indicator of sensitivity and specificity as continuous variables [42,43]. In this study, it was chosen to assess the accuracy of the predictions.

### 2.3.2. Calculation of Ecosystem Vulnerability

Ecosystem vulnerability refers to the sensitivity and resilience of an ecosystem to external disturbance. From the perspective of the landscape background, the degree of disturbance of landscape types, the landscape vulnerability index, and the landscape adaptive capacity index can reflect the vulnerability of ecosystems. The ecosystem vulnerability index is as follows [40]:

$$LVI = U \times V / (1 + LAI)$$

where $LVI$ represents ecosystem vulnerability, $U$ represents the degree of disturbance of landscape types, $V$ represents landscape vulnerability, and $LAI$ represents the landscape adaptability index (Table 1).

### 2.3.3. Calculation of Ecosystem Exposure

In this study, the value of ecosystem services is considered as the receptor of ecological risk. Based on the equivalent table of the value of ecosystem services per unit area in

China estimated by Xie et al. (2015) [44] through a survey of 500 ecologists, it is adjusted in combination with the land use types in the Qilian Mountains. Therefore, the ecosystem service value formula is as follows:

$$E_i = \sum_{j=1}^{9} k_{ij} A_j$$

$E_i$ represents the value of ecosystem services for a certain landscape type *i*. $k_{ij}$ represents the value-weighted factor of *j* types of ecosystem service functions for landscape type *i*. $A_j$ represents the area of landscape type *i*.

**Table 1.** Indicators used to assess ecosystem vulnerability.

| Indicator | Calculation | Landscape Ecology Significance |
|---|---|---|
| Interference degree of landscape (*U*) | $U = a \times FN + b \times FI + c \times FD$ | The degree of landscape disturbance indicates the degree of loss in the area after being disturbed. Based on expert opinions, take values of 0.5, 0.3, and 0.2 for *a*, *b*, and *c*, respectively. |
| Fragmentation (*FN*) | $FN = MPS \times (N_f - 1)/N_c$ | The degree to which the landscape is fragmented and broken reflects the extent of human interference with the landscape to a certain extent. |
| Isolation of landscape (*FI*) | $FI = \frac{1}{2}\sqrt{\frac{\pi}{A}} \times \frac{A_i}{A}$ | Used to reveal the degree of separation of individual patches in landscape types. The higher the detachment of patches, the smaller their ability to resist risks, and the lower their landscape safety. |
| Fractal dimension (*FD*) | $FD = 2\ln(P/4)/\ln A$ | The smaller the index value, the less likely it is to be interfered with by human activities, while the larger the index value, the more likely it is to be interfered with by human activities. |
| Landscape vulnerability (*V*) | Experts knowledge acquired | Barren, Sonw/Ice, Water, Wetland, Grassland, Forest, Shrub, Cropland, and Impervious are divided into 9, 8, 7, 6, 5, 4, 3, 2, and 1, respectively. |
| Landscape adaptation index (*LAI*) | $LAI = PRD + SHDI + SHEI$ | *LAI* is determined by referencing other people's research and is composed of the patch richness index, Shannon diversity index, and Shannon evenness index. |
| Patch richness index (*PRD*) | $PRD = m/A$ | *PRD* is the number of patches per unit area; it reflects the dispersion of patches in a landscape type. |
| Shannon diversity index (*SHDI*) | $SHDI = -\sum_{i=1}^{m}(p_i \times \ln p_i)$ | Reflects the richness and complexity of the landscape. |
| Shannon evenness index (*SHEI*) | $SHEI = \frac{\sum_{i=1}^{m}(p_i \times \ln p_i)}{-\ln m}$ | Indicates the maximum possible diversity of a given landscape's richness. Reflects the stability of the ecosystem. |

## 3. Results

### 3.1. Hazard Prediction Results

There were 2252 pixels in the training set and 965 pixels in the test set. The results show that when gamma is 0.5 and C is 5, the AUC value is relatively high with minimal loss of accuracy, making the model optimal (Figure 3). As shown in Figure 4, the AUC value for disaster assessment based on the SVM model reached 0.8898.

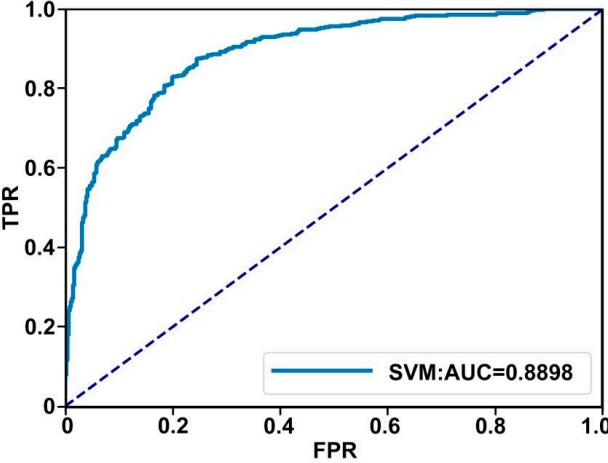

**Figure 4.** Accuracy verification.

*3.2. Results of ERA*

As shown in Figure 5a, the calculation results of the ERA are presented using a color gradient from red to blue, where red represents a higher ecological risk and blue represents a lower ecological risk. It can be seen that the red areas are concentrated in the eastern region and scattered in the western and peripheral areas. The forests and grasslands of the Qilian Mountains are mainly distributed in the east, where ecosystem services are concentrated, and the ecological service values are generally higher. In contrast, the western region is dominated by large areas of bare land, sandy areas, and deserts, resulting in sparse and fragmented ecological resources and relatively lower ecosystem service values compared to the east. Therefore, we consider that the eastern part of the Qilian Mountains has a higher risk from human activities, coupled with the concentration of ecological functions in forests and grasslands, resulting in a high aggregation of ecological risk. Meanwhile, the western part of the Qilian Mountains has a dispersed distribution of potential ecological risk due to the relative dispersion of ecological sources and the high vulnerability of sandy and desert areas.

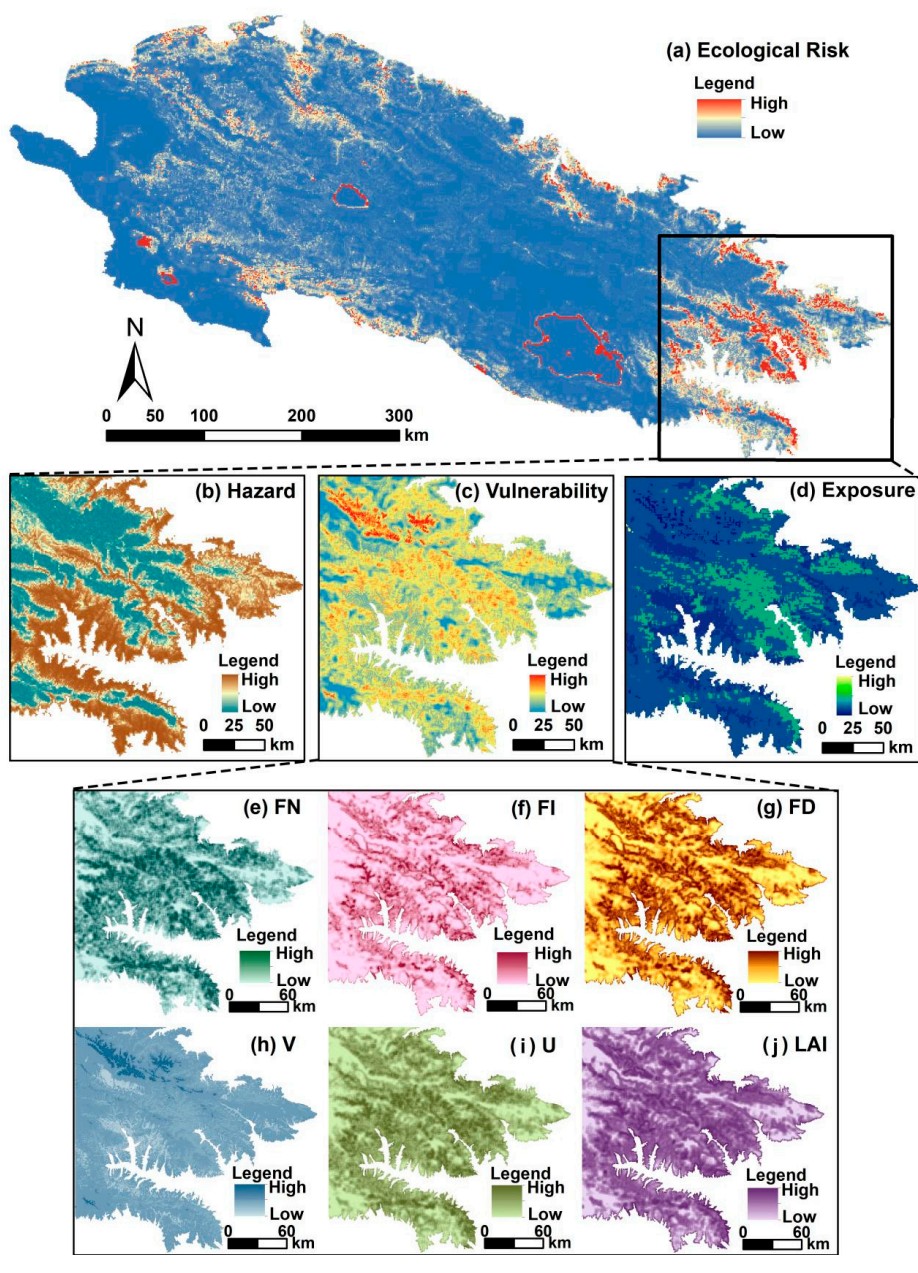

**Figure 5.** Results of ERA (see Table 1 for explanation of variables).

Taking the example of the eastern region with the highest risk, we analyze the indicators. The highest-risk areas are mainly in the low-lying river valleys of the eastern region, where vulnerability and exposure are significantly higher than in other areas. This area has a higher population density and more urbanization, with significant impacts from human activities such as deforestation, overgrazing, logging for fuelwood, and mining. Therefore, Figure 5e–g show higher values for landscape fragmentation, landscape fractal dimension, and isolation in this region. In addition, the predominant landscape types in the east are forests and grasslands, which generally have higher ecosystem service values, leading to higher vulnerability in this area.

### 3.3. Spatial Patterns of Ecological Risk

We used spatial analysis to discuss the spatial autocorrelation patterns of ecological risk and its assessment indicators. On a global scale, the Moran indices of ecological risk and hazard are 0.288 (significant at the $p = 0.05$ level) and 0.287 (significant at the $p = 0.05$ level), respectively, both showing an agglomeration distribution pattern. The Moran index of vulnerability and exposure are 0.079 and 0.124, respectively, but do not pass the significance test and both show a random distribution pattern. Figure 6a–d shows the local spatial autocorrelation patterns of various indicators of ecological risk, where ecological risk shows a high-value agglomeration in the eastern part of the Qilian Mountains and a low-value agglomeration in the central part. The spatial distribution of hazard is consistent with that of ecological risk. Vulnerability shows a high-value agglomeration in the eastern part of the Qilian Mountains and is not significant in other areas. Exposure shows a high-value agglomeration in the southern part of the Qilian Mountains and a low-value agglomeration in the western part.

Following the natural breakpoint method, the ecological risk and its three-dimensional assessment indicators are classified into low risk, medium risk, and high risk. With the ecological risk classification as the base map, the classification combination patterns of hazard, vulnerability, and exposure are shown. From Figure 6e, there are mainly six combination patterns: low hazard–low vulnerability–low exposure, low hazard–low vulnerability–high exposure, low hazard–high vulnerability–low exposure, low hazard–high vulnerability–high exposure, high hazard–low vulnerability–low exposure, and high hazard–high vulnerability–low exposure. Table 2 summarizes the spatial combination patterns distributed in different levels of ecological risk. In low ecological risk, low hazard–low vulnerability–low exposure is the main combination pattern, accounting for 50%. In medium ecological risk, high hazard–high vulnerability–low exposure is the main combination pattern, accounting for 46.2%, followed by low hazard–high vulnerability–low exposure, accounting for 30.8%. At high ecological risk, high hazard–high vulnerability–low exposure is the main combination pattern, accounting for 83.3%.

**Table 2.** Statistics on the ecological risk of geological disasters and the combination patterns.

| Combination Pattern | Low Risk | | Medium Risk | | High Risk | |
|---|---|---|---|---|---|---|
| | Functional Unit/Number | Share/% | Functional Unit/Number | Share/% | Functional Unit/Number | Share/% |
| Low hazard–low vulnerability–low exposure | 8 | 50 | 0 | 0 | 0 | 0 |
| Low hazard–low vulnerability–high exposure | 0 | 0 | 1 | 7.7 | 0 | 0 |
| Low hazard–high vulnerability–low exposure | 4 | 25 | 4 | 30.8 | 1 | 16.7 |
| Low hazard–high vulnerability–high exposure | 1 | 6.3 | 0 | 0 | 0 | 0 |
| High hazard–low vulnerability–low exposure | 3 | 18.8 | 2 | 15.4 | 0 | 0 |
| High hazard–high vulnerability–low exposure | 0 | 0 | 6 | 46.2 | 5 | 83.3 |
| Total | 16 | 100 | 13 | 100 | 6 | 100 |

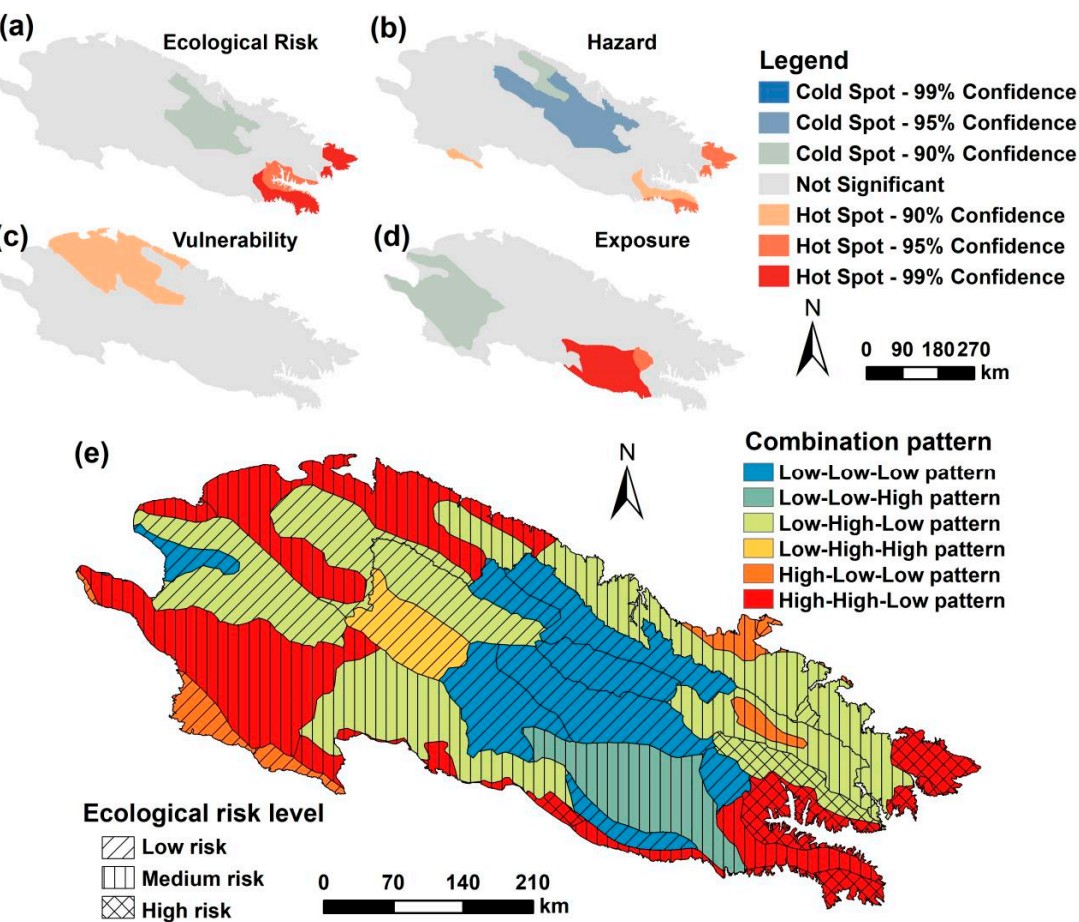

**Figure 6.** (**a**–**d**) Spatial hot and cold spot analysis of geological disaster ecological risk, hazard, vulnerability, and exposure. (**e**) Combination pattern of geological disaster ecological risk.

*3.4. Sources of Ecological Risk*

We performed a linear fit of the risk values of ecosystem functions with geological disaster hazard, vulnerability, and exposure, respectively (Figure 7a), with $R^2$ values of 0.37, 0.19, and 0.17, respectively, passing the significance test. This indicates that ecological risk is positively correlated with geological disaster hazard, vulnerability, and exposure overall, and is more affected by geological disaster hazard. To further reveal the causal mechanism of geological disaster ecological risk in the Qilian Mountains, we selected population density and road density as social factors and elevation and slope as natural environmental factors, combined with the three indicators of ecological risk to construct an SEM model of influencing factors and ecological risk. As shown in Figure 7c, the path coefficients of hazard, vulnerability, and exposure with ecological risk are 0.802 (significant at the $p = 0.01$ level), 0.438 (significant at the $p = 0.05$ level), and 0.286 (significant at the $p = 0.01$ level), respectively. From the path coefficients of social factors and natural environmental factors with hazard, vulnerability, and exposure, the path coefficient of hazard with social factors is higher than that with natural factors, indicating that geological disasters in the Qilian Mountains are greatly affected by human activities. The path coefficient of vulnerability with social factors is 5.275 (significant at the $p = 0.1$ level), while its path coefficient with natural environmental factors is 1.269 (significant at the $p = 0.1$ level), indicating that vulnerability is jointly affected by human activities and natural environmental factors. The path coefficient of exposure with social factors is also higher than that with natural factors. Therefore, we believe that geological disaster hazards affected by human activities are the main source of ecological risk, followed by vulnerability distribution affected by both human activities and the natural environment as a secondary source of ecological risk.

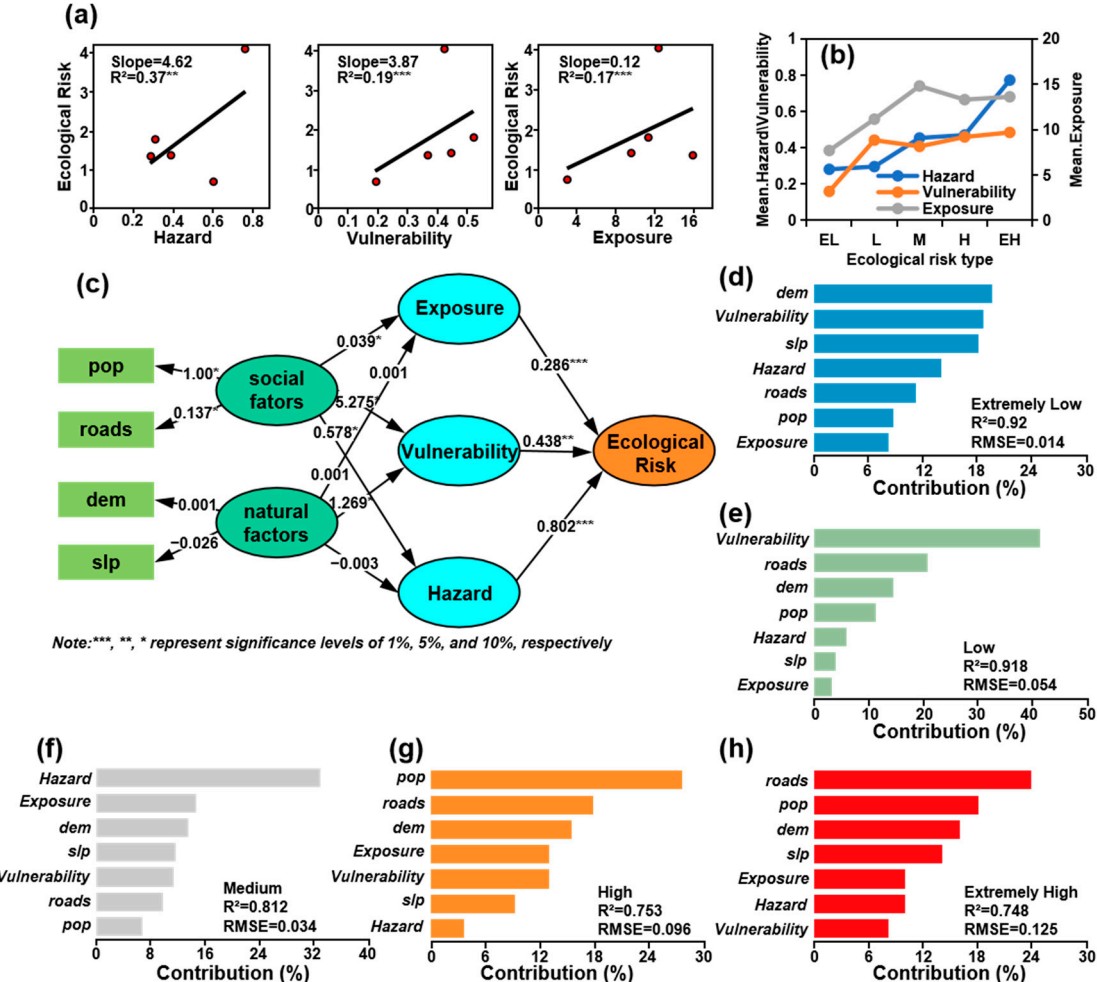

**Figure 7.** Identification of ecological risk sources. (**a**) the correlation between hazard, vulnerability, and exposure; (**b**) the average value of hazard, vulnerability, and exposure of different ecological risk levels. (**c**) SEM of Ecological Risk. (**d**–**h**) the contribution of different environmental factors to ecological risk.

We divided the geological disaster ecological risk into five levels: extremely low, low, medium, high, and extremely high. Using the random forest model, we calculated the importance of influencing factors at each level. From Figure 7d–h, it can be seen that in the extremely low ecological risk zone, the contribution of natural environmental factors and vulnerability is relatively high, both exceeding 18%. In low-risk zones, the contribution rate of vulnerability is significantly higher than other factors, exceeding 40%. In medium-risk zones, the contribution of hazard is the highest. In high- and extremely high-risk zones, the contribution of social factors such as human density and road density is significantly higher than other factors. The importance of factors calculated by the random forest model is consistent with the results of the SEM model.

## 4. Discussion

### 4.1. Spatial Policies for Addressing Ecological Risks of Geological Disasters

Based on the calculation results of ecological risk levels, spatial combination patterns, and source factors in the Qilian Mountains, a total of 35 functional units were grouped into four clusters using a systematic clustering method to formulate policy zones for managing geological hazard risks. Cluster 1 includes 15 ecological functional units located in the outer ring of the Qilian Mountains, which mainly function in water conservation and biodiversity. Cluster 2, located at the outermost periphery, consists of six ecological functional units,

mainly focusing on soil conservation. Cluster 3 consists of 13 ecological functional units located in the eastern and central parts, which are mainly dedicated to biodiversity protection. Cluster 4 consists of one ecological functional unit. The distribution of ecological risk results across the different clusters shows a clear concentric structure (Figure 8). Cluster 4, which is the innermost cluster, has the lowest risk, followed by clusters 3 and 1, while cluster 2, which is the outermost cluster, has the highest ecological risk.

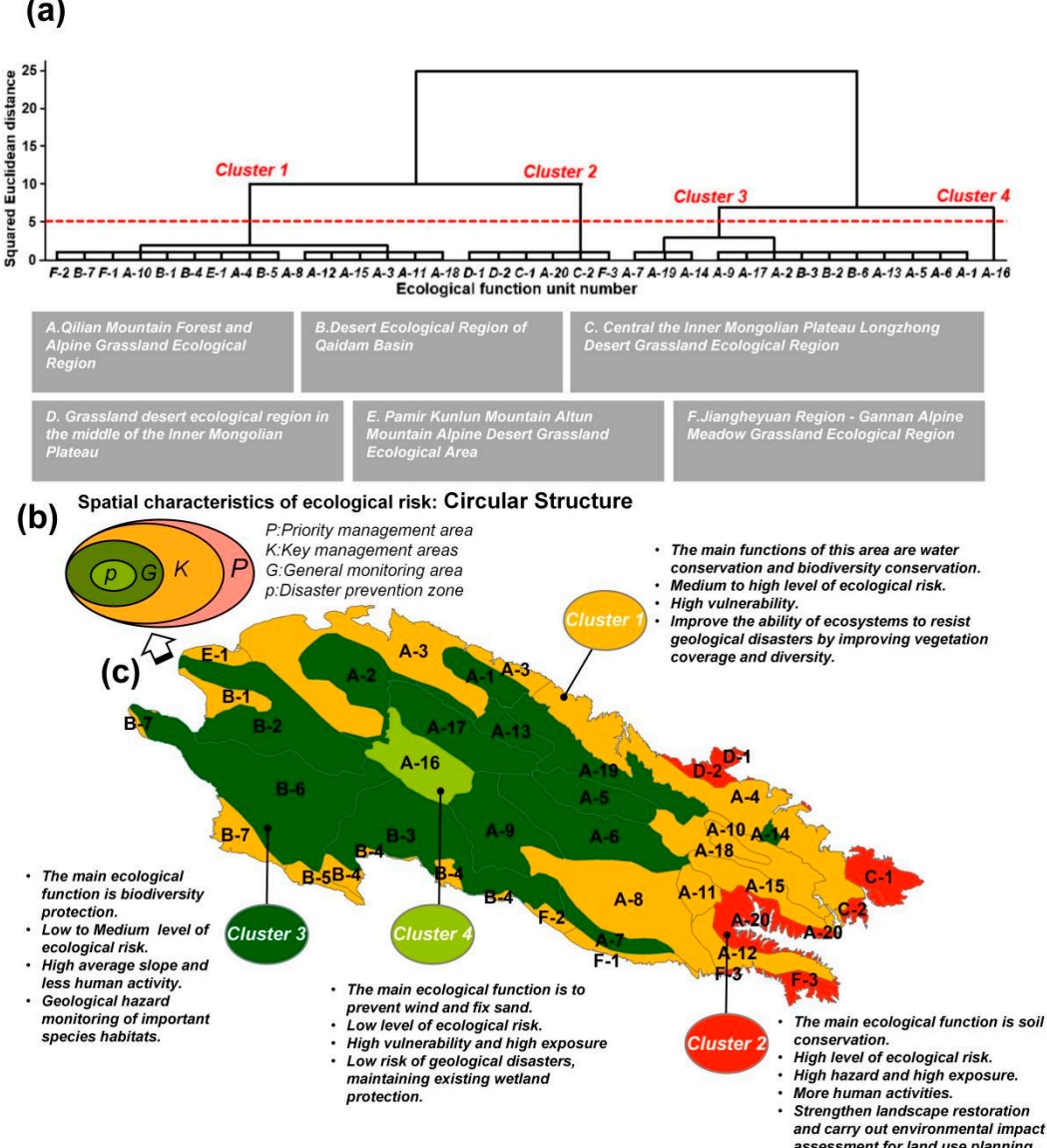

**Figure 8.** Spatial strategies for addressing ecological risks of geological disasters. (**a**) systematic clustering of ecological functional units; (**b**) spatial characteristics of ecological risk; (**c**) space policy map.

Cluster 2 areas are characterized by high risk levels and high exposure, making them priority areas for hazard prevention and management. Since this region is also a primary area for human activities, we suggest strengthening geological hazard monitoring [35] and enhancing landscape restoration, coupled with conducting environmental impact assessments for land use planning.

Compared to Cluster 2, Cluster 1 exhibits relatively higher average vulnerability but lower hazard and exposure levels. Additionally, this region shows a belt-like distribution, where potential ecological risks from geological hazards directly impact ecological processes such as species migration and population distribution among ecological source areas [45]. In

geological hazard prevention and control, enhancing ecosystem resilience can be achieved through measures such as improving vegetation coverage and diversity [46].

In cluster 3 areas, geological hazard ecological risks primarily stem from natural environments. Compared to Clusters 1 and 2, these areas experience relatively low precipitation, sparse populations, economic underdevelopment, fewer objects threatened by disasters, and predominantly desert landscapes. Focus in this region should be on protecting mountain vegetation, controlling soil erosion, enhancing local soil and water conservation capacities, and strengthening geological hazard risk monitoring for important species habitats.

Cluster 4 areas exhibit relatively low potential ecological risks from geological hazards. Due to the low geological hazard risks and good ecological resilience foundation, prevention of ecological risks takes precedence in this region.

### 4.2. Advantages of the Technical Framework Proposed in This Study

Natural disasters can cause changes in the quality of the ecological environment, resulting in some damage to ecosystems [47]. The management of ecosystems under disaster risk pressure has become an increasingly pressing issue [48,49]. Effective maintenance of existing healthy ecosystems and comprehensive management and restoration of degraded ecosystems have become critical issues that require urgent attention [50,51]. In this study, we developed a quantitative analytical framework to assess regional ecological security based on the causal logic between geological hazard disturbances and ecosystems. We assigned the results of geological hazard ERAs to EFAs, thereby effectively identifying and quantifying the risk characteristics of different types of EFAs, and providing scientifically based optimization solutions. Compared to previous studies (2, 3, 31), the technical framework proposed in this study establishes links between ERA and EFAs (Figure 9), which can provide references for decision-making in ecosystem protection and management of EFAs under geological hazard pressure.

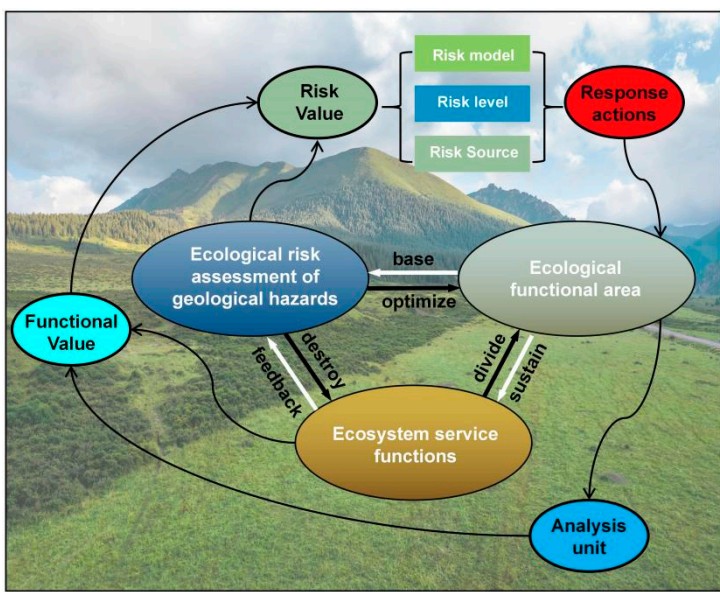

**Figure 9.** Framework for establishing the connection between ERA and EFAs.

### 4.3. Limitations and Prospects

This study establishes a technical framework for ecosystem risk management by linking geological disaster ERA with EFAs, which can help to better understand the safety status of EFAs and formulate targeted restoration and repair plans according to the degree of risk, thus providing a scientific basis for geological disaster prevention and ecological restoration. However, this study also has some limitations. Disaster risk is mainly focused

on geological disaster risk. In the future, multiple indicators and complex models can be integrated to introduce more risk indicators related to ecological security, such as drought and flood, and to consider more factors related to risk management, such as climate change and socio-economics. More complex models that truly reflect the dynamic changes and mechanisms of ecosystems under risk disturbance can be used.

## 5. Conclusions

In this study, we conducted a comprehensive assessment of ecological risks associated with geological disasters in the Qilian Mountains region. Using a support vector machine model for hazard prediction, we determined optimal parameter configurations and delineated the spatial distribution patterns of ecological risks using geographic information systems. Using structural equation modeling and spatial analysis, we elucidated the mechanisms through which factors such as geological hazards, vulnerability, and exposure influence ecological risks. We also proposed spatial policies and management strategies tailored to different risk zones and discussed the advantages and limitations of our technical framework. Our research provides a scientific basis for ecosystem protection and management while advocating for future studies to deepen our understanding of ecological security through multi-factor, multi-model approaches to better address environmental challenges.

**Author Contributions:** X.L.: writing—original draft preparation, writing—review and editing, and project administration. Q.X.: methodology, software, investigation, data curation, conceptualization, validation and resources. H.H.: formal analysis, visualization and supervision. R.Z.: funding acquisition. All authors have read and agreed to the published version of the manuscript.

**Funding:** This research was funded by the National Natural Science Foundation of China (grant no. 42177466).

**Institutional Review Board Statement:** Not applicable, because this article does not contain any studies with human or animal subjects.

**Informed Consent Statement:** Not applicable.

**Data Availability Statement:** The datasets used and/or analyzed during the current study are available from the corresponding author upon reasonable request.

**Conflicts of Interest:** The authors declare that they have no known competing financial interests or personal relationships that could have appeared to influence the work reported in this paper.

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
