# Peer review of "Incorporating the Results of Geological Disaster Ecological Risk Assessment into Spatial Policies for Ecological Functional Areas: Practice in the Qilian Mountains of China"

_sustainability, doi:10.3390/su16072976_

Round 1
Reviewer 1 Report
Comments and Suggestions for Authors
I recently reviewed your manuscript titled "Incorporating the Results of Geological Disaster Ecological Risk Assessment into Spatial Policies for Ecological Functional Zones: Practice in the Qilian Mountains of China." Your work contributes valuable insights into integrating geological disaster risk assessments with spatial policies, which is essential for protecting these vulnerable ecosystems against the backdrop of climate change and human activities.
The detailed comments and suggestions have been annotated directly within the manuscript for your convenience. These encompass both specific technical points and broader recommendations on how to enhance the overall clarity and coherence of your argumentation.

Comments on the Quality of English LanguageMust be improved
Author Response
Thank for your comments. We revised our manuscript according to the comments. All major changes are red-marked in the revised manuscript.

Reviewer 2 Report
Comments and Suggestions for Authors
lines 44-74: some parts of introduction are trivial, consider reducing this paragraph, as the paper is actually starting at line 75
line 120: and are approximately
line 137: it would be good to add additional explicit information describing origin of data sources, like remote sensing data, census data, etc, e.g. data source #1 looks to be based on something similar to Landsat satellite
line 139: the choice of the nearest neighbor technique possibly needs to be proven as for layers with 30 m resolution such technique means random choice out of ~1000 values
line 200: Additional brief information on how the survey was conducted will improve understanding of the values inside the table 3.
line 206: adjust table width so that entries for forest, shrub, water etc will be shown in a single row, explain meaning of negative value, explain the meaning of impervious, and sonwice
lines 210-211: Description of transfer from grid scale to unit scale on fig. 6 is absent
line 224: Fig. 7(e-g)
line 245: Consider adding (see table 2 for explanation of variables)
line 253: consider changing to condensed
lines 255-256: No visible evidence for such conclusion is seen on figure 8, e.g. climate regulation or waste disposal have larger share of high values
line 262: Change to "We are using...."
line 266: Consider changing to do not pass
line 279: Table 4
line 284: Check using upper - lower cases for Low and High
lines 381, 392: consider changing to present tense - is establishing....
Comments on the Quality of English Language
Additional check of the grammar is suggested, like in line 253 (consider changing to condensed), line 262 (consider changing "We are using...."), line 266 (Consider changing to do not pass...), lines 381, 392 (consider changing to present tense - is establishing....) etc
Author Response

(The authors gave the same response as above.)

Reviewer 3 Report
Comments and Suggestions for Authors
The authors proposed the following manuscript: "Incorporating the Results of Geological Disaster Ecological Risk Assessment into Spatial Policies for Ecological Functional Zones: Practice in the Qilian Mountains of China".
The manuscript is very well structured and contains much useful data in establishing the technical framework for environmental risk assessment of geological disasters based on machine learning, geographic information technology and multivariate statistical modelling analysis.
The main research conclusion is that the social factors represents the main source of geological disaster ecological risk in the studied area-Qilian Mountains.
The technical framework proposed in this study establishes the connection between geological disasters, ecological functional areas, and ecosystem service functions, providing a reference for decision-making on ecosystem protection and management in ecological functional areas under geological disaster stress.
In consideration of the above, I propose the manuscript for publication.
Author Response
感谢您的评论。我们根据评论修改了我们的手稿。所有重大改动均在修订稿中标有红色标记。
